# Psychiatric inpatient care for persons with dissociative identity disorder: a scoping review protocol

Anja Söderberg ,[1] Britt-Marie Lindgren,[2] Git-Marie Ejneborn Looi,[1] Josefin Bäckström ,[1,3] Sebastian Gabrielsson [1]

¹Department of Health, Education and Technology, Luleå University of Technology, Luleå, Sweden
²Department of Nursing, Umeå University, Umeå, Sweden
³Department of Medical Sciences, Psychiatry, Uppsala University, Uppsala, Sweden

**Correspondence to**
Anja Söderberg;
anja.soderberg@ltu.se

## ABSTRACT

**Introduction** Psychiatric inpatient care (PIC) is often characterised by high pressure and thresholds for admission, brief periods of care and limited time for caring activities. Dissociative identity disorder (DID) is a contested diagnosis, and persons with DID are at risk of not receiving adequate support when cared for in PIC. Because the limited literature addressing the topic includes no overview on how persons with DID are cared for in psychiatric inpatient settings, the aim of this scoping review is to map the area of knowledge on PIC for persons experiencing DID. This scoping review will provide an overview with the possibility to elucidate gaps in the evidence base and needs for future research on PIC for persons experiencing DID.

**Methods and analysis** This scoping review will follow Preferred Reporting Items for Systematic Review and Meta-Analysis for Scoping Reviews and steps 1–5 described in the established method for scoping reviews: identifying research question, identifying relevant studies, study selection, charting the data and collating, summarising and reporting results.

**Ethics approval** Not applicable.

**Dissemination** This scoping review will be submitted for publication in an international, peer-reviewed journal.

## INTRODUCTION

Persons with experience of dissociative identity disorder (DID) and other severe dissociative states following trauma experience represent a vulnerable group in psychiatric inpatient care (PIC). This calls for contemporary, person-centred approaches to care that prevent retraumatisation and further harm.[1] In those efforts, awareness of the importance of trauma-informed care (TIC) in psychiatric inpatient settings has increasingly been raised.[2–6] The focus for this proposed scoping review is to provide an overview of what is known about PIC for persons experiencing DID.

## BACKGROUND

PIC is a form of care in which the person is cared for 24 hours a day and the daily activities

## STRENGTHS AND LIMITATIONS OF THIS STUDY

⇒ This proposed scoping review aims to map the area of knowledge broadly, encompassing various types of publications.
⇒ Quality assessment will be integral to the data charting process, contributing to its strength.
⇒ A limitation of this review is that it will not synthesise the findings.

are regulated by staff.[7] Following the widespread deinstitutionalisation in the second half of the 20th century, major changes have been made in the organisation of psychiatric care around the world,[8] such that PIC is no longer the dominant form of care.[7] Access to PIC varies worldwide, from <2 to >25 hospital beds per 100 000 population.[9] McCrae[10] argues that along with the organisational changes, the view on PIC has also shifted to the idea that PIC in itself is something bad that should be avoided at all costs. Beyond that, PIC settings are often characterised by high pressure and thresholds for admission, brief periods of care and limited time for caring activities.[10] Even so, PIC is considered to be necessary for persons with acute psychiatric problems,[11] and PIC of good quality is described to be based on relationships and moreover to be caring, person centred and recovery oriented.[12] Despite recommended models, including TIC, developed to improve the quality of care in PIC,[2–6] PIC facilities struggle to embrace a coherent vision and goal for the delivery of care.[13]

Dissociative states, among which DID is considered to be the most severe type,[14] include symptoms of depersonalisation, derealisation,[15] flashbacks, nightmares and switching between separate parts of the identity through compartmentalisation.[16] DID is defined as a complex yet valid diagnosis that can be distinguished from other disorders by assessing identity alteration and amnesia and

**Table 1** Search strategy

| PCC | PubMed | Cinahl | PsycINFO |
|---|---|---|---|
| Population: DID | (dissociative identity disorder (MeSH) OR 'multiple personality disorder') | ('dissociative identity disorder' OR multiple personality disorder (MH) OR did OR mpd) | (dissociative identity disorder (DE) OR 'multiple personality disorder') |
| Concept: psychiatry | (psychiatry (MeSH) OR psychiatric OR mental health (MeSH) OR 'mental illness' OR mental disorders (MeSH)) | (psychiatry (MH) OR psychiatric OR mental health OR 'mental illness' OR mental disorders (MH)) | (psychiatry (DE) OR psychiatric OR mental health (DE) or 'mental illness' (DE) OR mental disorders (DE)) |
| Context: inpatient care | (inpatients (MeSH) OR ward OR hospital* OR 'acute setting') | (inpatients (MH) OR ward OR hospital* OR 'acute setting') | (inpatients OR ward OR hospital* OR 'acute setting') |

is typically associated with severe childhood trauma.[17] Stigma is recognised to affect persons with dissociative states, who may be undiagnosed for several years and feel shame about their experiences of dissociation.[14] DID is also known to be a controversial and questioned diagnosis[14 16 18 19] with symptoms that can be misinterpreted as better known diagnosis such as schizophrenia or personality syndrome,[14 16 19] and the person with DID is at risk of not getting adequate treatment and being met with disbelief[1 19] and a lack of knowledge.[20]

Persons experiencing DID are at great risk of self-injury and suicide attempts[1 21–25] and may require PIC in order to ensure their personal safety and a stabilised well-being.[14] There is reason to believe that people with DID would benefit from contemporary developments of PIC, including person-centred, recovery-oriented and trauma-informed approaches to care.[1] To date, research on PIC for persons with DID is limited and without any

**Table 2** Pilot search

| Search | Search terms | Matches |
|---|---|---|
| S1 | exp dissociative identity disorder/ | 1186 |
| S2 | 'multiple personality disorder'.mp | 356 |
| S3 | S1 or S2 | 1218 |
| S4 | exp psychiatry/ | 760 502 |
| S5 | psychiatric.mp | 937 395 |
| S6 | exp mental health/ | 487 022 |
| S7 | 'mental illness'.mp | 40 185 |
| S8 | exp mental disorders/ | 1 459 534 |
| S9 | S4 or S5 or S6 or S7 or S8 | 2 079 336 |
| S10 | exp inpatients/ | 158 439 |
| S11 | ward.mp | 101 353 |
| S12 | hospital*.mp | 6 577 438 |
| S13 | 'acute setting'.mp | 2637 |
| S14 | S10 or S11 or S12 or S13 | 6 669 493 |
| S15 | S3 and S9 and S14 | 260 |
| S16 | S3 and S9 and S14* | 120 |

PubMed 230 104.
*Limitations: published between 2000 and 2022.

coherent description on the experiences and impact of PIC for persons with DID cared for in PIC. In response, this proposed scoping review will provide an overview with the possibility to elucidate gaps in the evidence base and needs for future research. Due to the relatively unexplored area, a descriptive design is motivated, and this proposed scoping review is aiming to map the area of knowledge on PIC for persons with DID, striving to be as comprehensive as possible in order to identify all relevant literature regardless of study design. The method of scoping review is suitable due to its ability to address broad research questions (RQs), map relevant literature and elucidate the field of interest.[26]

## METHODS AND ANALYSIS

This scoping review protocol follows the Preferred Reporting Items for Systematic Review and Meta-Analysis (PRISMA) Protocols checklist[27] and the proposed scoping review will follow PRISMA for Scoping Reviews.[28]

Step 1–5 in the established method for scoping reviews described by Arksey and O'Malley will be performed as described in what follows[26]:

### Identifying research questions

This proposed scoping review aims to map the area of knowledge on PIC for persons with DID. In specific, the review seeks to explore the following RQs:

► RQ1: How is DID conceptualised in the context of PIC?
► RQ2: Why are persons with DID subject to PIC?
► RQ3: What mechanisms of PIC for persons with DID are described?
► RQ4: What experiences of PIC for persons with DID are described?
► RQ5: What outcomes of PIC for persons with DID are described?

Mechanisms refer to key components of care, including interventions, therapies and treatments expected to influence the outcomes of PIC.

### Identifying relevant studies

To be as comprehensive as possible, the search strategy will include both peer-reviewed papers published in international scholarly journals indexed in curated databases

**Table 3** Inclusion and exclusion criteria

| | Inclusion | Exclusion |
|---|---|---|
| Persons with DID | ▶ Studies focused on persons with DID<br>▶ Studies with small samples of persons with DID but allow distinguishing persons with DID from the other population studied | ▶ Studies focused on persons with DID and a comorbidity |
| Psychiatry | ▶ Studies focused on psychiatric care | |
| Inpatient care | ▶ Studies focused on any context in which the person is cared for 24 hours a day<br>▶ Studies with small samples of inpatient care setting but allow distinguishing the inpatient care setting from other settings studied | |
| Year of publication | ▶ 2000–2023 | |
| Language | ▶ English<br>▶ Swedish<br>▶ Norwegian<br>▶ Danish | |
| Type of study | ▶ Any paper reporting an original study<br>▶ Internationally published studies<br>▶ Grey literature formed as dissertations, theses, conference publications, unpublished manuscripts and reports from governmental and non-governmental organisations reporting empirical studies | ▶ Editorials<br>▶ Discursive papers<br>▶ Literature reviews |

and grey literature.[26] As some information related to health RQs may only be found in grey literature, it is suggested that a broad definition of grey literature should be adopted.[29] For the purpose of this proposed review, grey literature refers to research not intended for publication, or not yet published, in international scholarly journals, that is, dissertations, theses, conference publications, unpublished manuscripts and reports from governmental or non-governmental organisations reporting empirical studies.

### Search strategy
The first author will perform all of the described steps of the search strategy.

#### Curated databases
Consultations with a librarian have been undertaken, and pilot searches of the literature have been performed in order to refine the search strategy. The databases PubMed, CINAHL and PsycINFO have been identified as being suitable for this proposed scoping review, related to the research area. Key concepts related to the aim and the RQs have also been identified and will form the search blocks. Key concepts are organised in accordance with the Population-Concept-Context (PCC) framework, a framework for scoping reviews recommended by the Joanna Briggs Institute.[30] In the search blocks, subject headings will be used together with related terms in free text, and the Boolean operators ('AND' and 'OR') will be used to narrow and expand the search. Each search in the databases will be documented with date, search terms, number of search matches, selected studies and limitations. The search strategy for each chosen database is described in table 1, followed by an example of a pilot search (table 2).

#### Additional sources
Additional searches will be performed manually in Google Search, Google Scholar and relevant databases that collect theses. Overlapping search strategies are necessary when aiming for breadth, and using these search engines increases the likelihood of finding up-to-date grey literature. However, this approach may generate a large number of search hits, and the searches may not be easily replicable.[29] Therefore, a systematic approach with defined search terms and a strategy for managing a large volume of search hits will be implemented. To further identify relevant studies that may not have been found by performing the search strategy in the electronic databases and the additional searches, the reference lists of selected studies will also be screened. Searches for studies that have cited the selected studies will additionally be performed. Key journals, identified through the selected studies, will be searched manually to identify studies that may have been missed in the searches in the electronic databases, in the screening of reference lists and in the citation searches. Last, searches through existing knowledge and networks related to the research area will be performed to find information about, for example, studies that have not yet been published.[26]

### Study selection
To determine the relevance of the studies found by performing the search strategy, the inclusion and exclusion criteria will be applied to all the studies found by performing the search strategy. The aim of the inclusion and exclusion criteria is to keep eligible studies and exclude irrelevant studies.[26] In this proposed scoping review, the general inclusion and exclusion criteria will consist of year of publication, language and type of study,

| Table 4 | Screening and charting template |
|---|---|
| **Step 1: title and abstract screening** | |
| ► Is this title and abstract written in English/Swedish/Norwegian/Danish? | Yes/no |
| ► Does it seemingly address DID in PIC? | Yes/no |
| **Step 2: full-text screening** | |
| ► Is there any reason this article should be excluded? | Yes/no |
| If yes: what is the reason for excluding? | 1. Not in English/Swedish/Norwegian/Danish<br>2. Not published between 2000 and 2023<br>3. Not focus on DID<br>4. Not focus on PIC<br>5. No full text available despite efforts to retrieve |
| If no: what is the data charting information? | Use the data charting table |
| ► Are there descriptions on how DID is conceptualised in the context of PIC (RQ1)? | Yes/no |
| If yes, describe how: | |
| ► Are there descriptions of why persons with DID are subject to PIC (RQ2)? | Yes/no |
| If yes, describe how: | |
| ► Are there descriptions of mechanisms of PIC for persons with DID (RQ3)? | Yes/no |
| If yes, describe how: | |
| ► Are there descriptions of experiences of PIC for persons with DID (RQ4)? | Yes/no |
| If yes, describe how: | |
| ► Are there descriptions of outcomes of PIC for persons with DID (RQ5)? | Yes/no |
| If yes, describe how: | |

whereas other inclusion and exclusion criteria are formulated in accordance with the RQs (table 3).

Regarding the year of publication, a longer period of time is chosen in order to gather enough data and has been considered in relation to the relatively unexplored area. The limit of the year 2000 has been set due to the organisation of PIC after the deinstitutionalisation. In addition to English, Swedish, Norwegian and Danish have been chosen for language considerations, as the authors are Swedish and are proficient in these Scandinavian languages.

To apply the inclusion and exclusion criteria to all of the identified studies, a systematic approach will be undertaken. The screening of all studies will follow a combined template for screening and charting (table 4).

The results of the searches from each database will be downloaded in the screening tool Rayyan to support a systematic screening. A first, selection of studies will be made through applying the inclusion criteria on the title and abstract of all the studies. The next step will include reading the studies in full in order to make a final decision on whether to include or exclude, which is required since abstracts cannot be expected to represent the full study or to grasp the full scope of the study.[26] Both steps of the study selection will include two authors independently screening the studies. If there are disagreements on whether to include or exclude, the two authors will discuss and try to reach a consensus, and if needed, a third author will be involved to resolve discrepancies. The process of selecting eligible studies will follow the PRISMA flow diagram (figure 1).[31]

### Charting the data

The combined template for screening and charting (table 4) will also include steps for charting the data and will be used to support a systematic approach. Data from the selected studies will then be charted according to the RQs. The charted data will be transferred to a data charting table (online supplemental table 1) that includes the following information:
► Authors
► Year of publication
► Country
► Population
► Setting
► Aim
► Design
► Discipline
► Quality
► RQ1
► RQ2
► RQ3
► RQ4
► RQ5

The charted data will then be used as the foundation for analysis.[26] Two authors will chart the data independently, and disagreements will be resolved through discussion until consensus is reached.

### Quality assessment

Because scoping reviews generally seek comprehensiveness, breadth and inclusion, a quality assessment is not included in the description of a scoping review which is acknowledged as a limitation of the method.[26] However, in this proposed scoping review, a quality assessment will be performed as a part of the data charting in order to also provide an overview of the quality of the existing

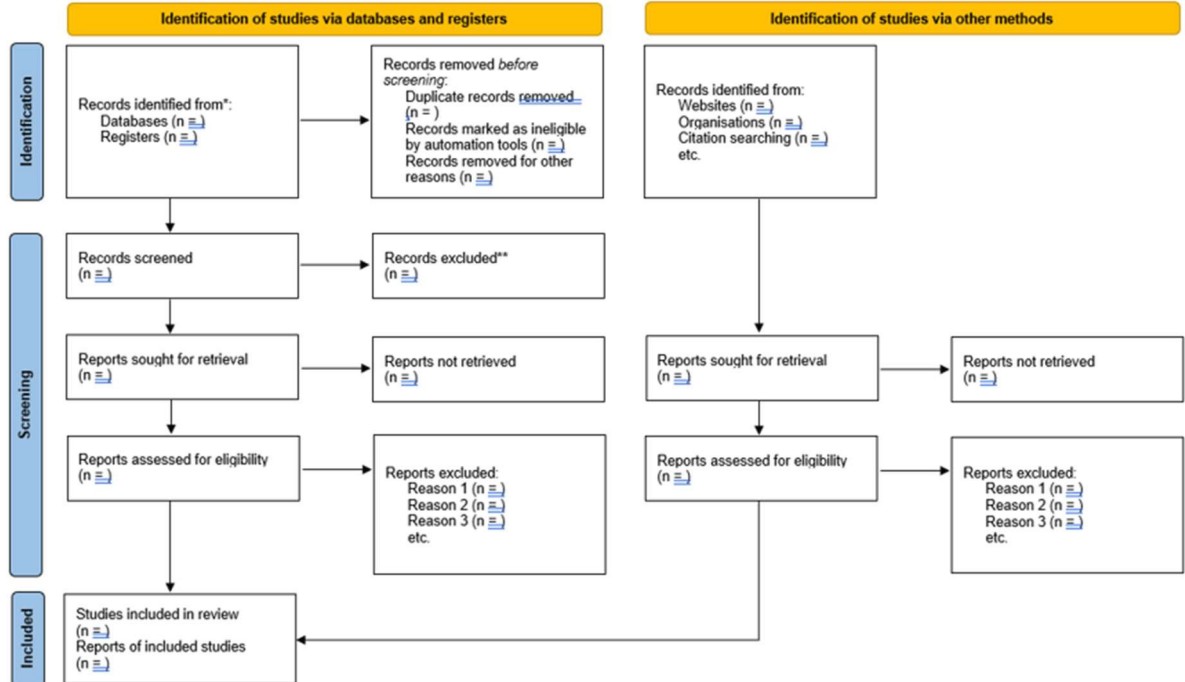

**Figure 1** PRISMA flow diagram.

research. The Mixed Method Appraisal Tool[32] will be used for the quality appraisal of the found studies. This tool is developed to be used in reviews containing qualitative, quantitative and mixed method studies,[32] which is why it is chosen for the quality appraisal of this review. Two authors will independently perform the quality appraisal, and disagreements will be discussed until a consensus is reached. A third author will be involved in the process to resolve discrepancies if needed.

## Collating, summarising and reporting the results

The final part of this proposed scoping review will provide an overview of the research by collating, summarising and reporting results. One part of this analysis will be a basic numerical analysis of the included studies, meaning the data presented in the data charting table (online supplemental table 1), in order to bring light to dominant areas of research by to country, study population, setting, aims, designs and disciplines. The second part of the analysis will be to organise the data thematically according to the RQs. A clear reporting strategy will be undertaken in order to determine potential bias, which is why a consistent approach based on the combined template for screening and charting (table 4), also covering the RQs, will be used. The next step will be to make comparisons, identify contradictions, identify gaps in the evidence base and suggest topics for future research.[26] All authors will be involved in the final discussion about the collated, summarised and reported results.

## Patient and public involvement

A non-governmental organisation advocating the interests of persons experiencing DID, including people with personal experience of DID, has been engaged in the development of the RQs and the design of the review. Members of the board were invited to take part in the research plan and participated in several discussions on the research plan together with the research team. No major changes were made after discussions with the non-governmental organisation. The findings from the review will be communicated with representatives of the organisation.

## ETHICS APPROVAL

Not applicable

## DISSEMINATION

This proposed scoping review will be submitted for publication in an international, peer-reviewed journal

**Correction notice** This article has been corrected since it was published. Licence updated to CC BY on 1st August 2024.

**Contributors** AS, SG, B-ML and G-MEL have made substantial contributions to the conception and the design. AS and SG have drafted the work, and AS, SG, B-ML, G-MEL and JB have substantively revised it. Each author has approved the submitted version, and all authors have agreed to be personally accountable and to ensure that questions related to the accuracy or integrity of any part of the work, even ones in which the author was not personally involved, are appropriately investigated and resolved, and the resolution is documented in the literature.

**Funding** This research is funded by Luleå University of Technology.

**Competing interests** None declared.

**Patient and public involvement** Patients and/or the public were involved in the design, or conduct, or reporting, or dissemination plans of this research. Refer to the Methods section for further details.

**Patient consent for publication** Not applicable.

**ORCID iDs**
Anja Söderberg http://orcid.org/0000-0001-7066-955X
Josefin Bäckström http://orcid.org/0000-0002-9440-1985
Sebastian Gabrielsson http://orcid.org/0000-0002-1624-1795

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
