## [Reviewer comments · BMJ Open]

ARTICLE DETAILS

TITLE (PROVISIONAL)	Psychiatric inpatient care for persons with dissociative identity disorder: A scoping review protocol
AUTHORS	Söderberg, Anja; Lindgren, Britt-Marie; Ejneborn-Looi, Git-Marie; Bäckström, Josefin; Gabrielsson, Sebastian

VERSION 1 – REVIEW

REVIEWER	Brand, Bethany L. Towson University
REVIEW RETURNED	03-Dec-2023

GENERAL COMMENTS	The authors have chosen an important topic to study. The review process is well-thought out and described. I have only minor suggestions. 1. The first time they use abbreviations such as "PIC", the full term should be used along with the abbreviation.2. They use "traumainformed" in some places and "trauma-informed" in other places. The latter is preferred.3. They have not cited some of the primary studies that show that dissociative identity disorder patients are misdiagnosed for years, that they are at high risk for suicide attempts and non-suicidal self-injury (e.g., Foote, Smolin et al. 2008; Brand et al., 2009, 2013, 2019), and that they face considerable treatment barriers including stigma (e.g., Leonard et al.; Nester et al; 2022) and lack of appropriately trained clinicians (e.g., Kumar et al). This absence of citations of key studies brings up the question as to whether the authors will be successful in finding all of the relevant research needed for a thorough review. However, as they outline their process, it seems they should be able to discover relevant studies.
---

REVIEWER	Rissanen, Marja-Liisa Tampere University
REVIEW RETURNED	11-Dec-2023

GENERAL COMMENTS	Bachelor's or master's theses should not be included in your literature search. Their level is not sufficient for scoping review.
---

REVIEWER	Machaczek, Katarzyna Sheffield Hallam University, Advanced Wellbeing Research Centre
REVIEW RETURNED	30-Dec-2023

GENERAL COMMENTS	REVISED COMMENTS:
-------------------

The aim of the proposed review is to answer a number of research questions that are linked to psychiatric inpatient care (PIC) for people with dissociative identity disorder (DID).

Please see below a number of comments and suggestions which, I hope, will help improve the protocol.

Abstract:

The use of the word 'proposed' in the abstract is redundant.

Introduction:

Dissociative identity disorder isn't defined, although some of the symptoms are listed.

Research question 3: 'What mechanisms of PIC for persons with DID are described?' Can the authors specify how they define 'mechanisms' for the purpose of their review?

Methods:

Grey literature:

The authors might want to consider additional references to the definition of grey literature. Please see, for example:

<https://systematicreviewsjournal.biomedcentral.com/articles/10.1186/s13643-016-0337-y>

Pilot searches of the literature:

The authors state that 'Consultations with a librarian have been undertaken and pilot studies have been performed in order to refine the search strategy.' Do they mean 'pilot searches of the literature'?

Established framework to aid review of literature:

To ensure that the searches are comprehensive and to reduce the risk of bias, the authors could consider using one of the established frameworks for aiding literature reviews, such as the population, intervention, comparator, outcome, and study type (PICOS) framework. This would aid the quality of

	the review. For example, the search strategy does not include patient outcomes although the authors state that they plan to extract this information from the studies included in the review. Additional search sources: The authors state that additional searches for literature will be performed using Google, Google Scholar and 'possibly other databases as well'. It would be useful if they could add more details about the additional sources. If not planned effectively, website searching can introduce bias. It would be useful to see the section that considers the strengths and limitations of each additional search resource. It might be useful to add searches on online depositories, library catalogues, etc., in addition to Google and Google Scholar – given that the authors' intention is to identify dissertations and theses. Do the authors plan to use any software for managing literature reviews, such as COVIDENCE, to aid the review process? Quality assessment: The authors state that they intend to use the mixed methods appraisal tool to assess the quality of literature included in the review. What about the quality assessment of grey literature? The authors might want to consider using one of the quality assessment tools that have been developed specifically for grey literature – and which are likely to be more appropriate than the mixed methods appraisal tool. Languages other than English (LOE): Including papers in LOE in the review is commendable. Could the authors please state explicitly in their protocol how they are going to search, screen and analyse LOE papers? There is an emerging body of literature that provides strategies for screening LOE titles and abstracts, for example. Patient and Public Involvement: In the patient and public involvement section, the authors state that non-governmental organisations have been involved in the development of the review questions. Could the authors please provide more details about this? The scoping review methodology allows for the involvement of key stakeholders, including people living with specific health conditions. Given
--	--

	the topic of the review, it would be extremely valuable if the authors considered the involvement of service users and their carers and healthcare providers.
--	---

VERSION 1 – AUTHOR RESPONSE

Reviewer: 1

Comments to the Author:

The authors have chosen an important topic to study. The review process is well-thought out and described. I have only minor suggestions.

- 1. The first time they use abbreviations such as "PIC", the full term should be used along with the abbreviation.**

Thank you for noticing this. We have now made sure that all abbreviations are defined at first mention.

- 2. They use "traumainformed" in some places and "trauma-informed" in other places. The latter is preferred.**

Thank you for noticing this, we have decided to use "trauma-informed" as suggested.

- 3. They have not cited some of the primary studies that show that dissociative identity disorder patients are misdiagnosed for years, that they are at high risk for suicide attempts and non-suicidal self-injury (e.g., Foote, Smolin et al. 2008; Brand et al., 2009, 2013, 2019), and that they face considerable treatment barriers including stigma (e.g., Leonard et al.; Nester et al; 2022) and lack of appropriately trained clinicians (e.g., Kumar et al). This absence of citations of key studies brings up the question as to whether the authors will be successful in finding all of the relevant research needed for a thorough review. However, as they outline their process, it seems they should be able to discover relevant studies.**

Thank you for this valuable comment and suggestions! We were familiar with some of the suggested studies, but the ones from 2022 (Kumar et al. and Nester et al.) were new to us! We agree, these key studies add value to the background and we have thus added them on page 4:

“DID is also known to be a controversial and questioned diagnosis [14, 16, 18, 19] with symptoms that can be misinterpreted as better known diagnosis such as schizophrenia or personality syndrome [14, 16, 19] and the person with DID is at risk of not getting adequate treatment and being met with disbelief [1, 19] and a lack of knowledge [20]. Persons experiencing DID are at great risk of self injury and suicide attempts [1, 21, 22, 23, 24, 25]”

20. Kumar, S., Brand, B. & Courtois, C. The need for trauma training: clinicians' reactions to training on complex trauma. *Psychol Trauma*. 2022;14(8): 1387-1394.
<https://doi.org/10.1037/tra0000515>

21. Brand, B., Classen, C., Lanins, R., Loewenstein, R., McNary, R., Pain, C. & Putnam, F. A naturalistic study of dissociative identity disorder and dissociative disorder not otherwise specified patients treated by community clinicians. *Psychol Trauma*. 2009;1(2): 153-171. <https://doi.org/10.1037/a0016210>

22. Brand, B., McNary, S., Myrick, A., Classen, C., Lanius, R., Loewenstein, R., Pain, C. & Putnam, F. A longitudinal naturalistic study of patients with dissociative disorders treated by community clinicians. *Psychol Trauma*. 2013;5(4): 301-308. <https://doi.org/10.1037/a0027654>

23. Brand, B., Schielke, H., Putnam, K., Putnam, F., Loewenstein, R., Myrick, A., Jepsen, E., Langeland, W., Steele, K., Classen, C. & Lanius, R. An online educational program for individuals with dissociative disorders and their clinicians: 1-Year and 2-year follow-up. *J Trauma Stress*. 2019; 32: 156-166. <https://doi.org/10.1002/jts.22370>

24. Foote, B., Smolin, Y., Neft, D.I. & Lipshitz, D. Dissociative disorders and suicidality in psychiatric outpatients. *J Nerv Ment Dis*. 2008;196(1): 29-36. <https://doi.org/10.1097/NMD.0b013e31815fa4e7>

25. Nester, M., Brand, B., Schielke, H. & Kumar, S. An examination of the relations between emotion dysregulation, dissociation, and self-injury among dissociative disorder patients. *Eur J Psychotraumatol*. 2022;13(1). <https://doi.org/10.1080/20008198.2022.2031592>

Reviewer: 2

Comments to the Author:

Bachelor's or master's theses should not be included in your literature search. Their level is not sufficient for scoping review.

Thank you for your comment! We will include master's theses but not bachelor's theses. Grey literature is considered to make important contributions to reviews and includes academic papers such as theses (see Paez, 2017, doi: 10.1111/jebm.12265)

Reviewer: 3

Comments to the Author:

REVISED COMMENTS:

The aim of the proposed review is to answer a number of research questions that are linked to psychiatric inpatient care (PIC) for people with dissociative identity disorder (DID).

Please see below a number of comments and suggestions which, I hope, will help improve the protocol.

Abstract:

1. The use of the word 'proposed' in the abstract is redundant.

Thank you for this suggestion. We have now removed the word "proposed" from the abstract on page 2.

Introduction:

2. Dissociative identity disorder isn't defined, although some of the symptoms are listed.

Thank you for noticing this, we agree that dissociative identity disorder needs to be further defined and we have added this on page 4:

"DID is defined as a complex yet valid diagnosis that can be distinguished from other disorders by assessing identity alteration and amnesia, and is typically associated with severe childhood trauma [17]."

3. Research question 3: 'What mechanisms of PIC for persons with DID are described?' Can the authors specify how they define 'mechanisms' for the purpose of their review?

We realise that "mechanisms" need to be further defined and we have added this on page 6:

"Mechanisms refer to key components of care, including interventions, therapies and treatments expected to influence the outcomes of PIC."

Methods:

Grey literature:

4. The authors might want to consider additional references to the definition of grey literature. Please see, for example:

<https://systematicreviewsjournal.biomedcentral.com/articles/10.1186/s13643-016-0337-y>

Thank you for this suggestion, we were not familiar with this article before and we agree that it adds value to the descriptions of grey literature. We have added this on page 6:

“As some information related to health research questions may only be found in grey literature it is suggested that a broad definition of grey literature should be adopted [29].”

Pilot searches of the literature:

- 5. The authors state that ‘Consultations with a librarian have been undertaken and pilot studies have been performed in order to refine the search strategy.’ Do they mean ‘pilot searches of the literature’?**

Thank you for noticing this mistake! We mean pilot searches which has now been clarified on page 6:

“Consultations with a librarian have been undertaken and pilot searches of the literature have been performed in order to refine the search strategy.”

Established framework to aid review of literature:

- 6. To ensure that the searches are comprehensive and to reduce the risk of bias, the authors could consider using one of the established frameworks for aiding literature reviews, such as the population, intervention, comparator, outcome, and study type (PICOS) framework. This would aid the quality of the review. For example, the search strategy does not include patient outcomes although the authors state that they plan to extract this information from the studies included in the review.**

We agree that an established framework will improve the quality of the review! We have organized the key concepts according to the PCC framework instead of PICOS since it is recommended for scoping reviews by the Joanna Briggs Institute. This is now clarified on page 6-7:

“Key concepts are organized in accordance with the Population-Concept-Context-framework (PCC), a framework for scoping reviews recommended by the Joanna Briggs Institute [30].”

We have also clarified this in table 1 on page 7:

Table 1: Search strategy			
PCC	PubMed	Cinahl	PsycINFO
Population: DID	(dissociative identity disorder [MeSH] OR “multiple personality disorder”)	(“dissociative identity disorder” OR multiple personality disorder [MH] OR did OR mpd)	(dissociative identity disorder [DE] OR “multiple personality disorder”)
Concept: Psychiatry	(psychiatry [MeSH] OR psychiatric OR mental health [MeSH] OR “mental illness” OR mental disorders [MeSH])	(psychiatry [MH] OR psychiatric OR mental health OR “mental illness” OR mental disorders [MH])	(psychiatry [DE] OR psychiatric OR mental health [DE] or “mental illness” [DE] OR mental disorders [DE])
Context: Inpatient care	(inpatients [MeSH] OR ward OR hospital* OR “acute setting”)	(inpatients [MH] OR ward OR hospital* OR “acute setting”)	(inpatients OR ward OR hospital* OR “acute setting”)

When it comes to outcomes, it is one aspect that is covered by the research questions. When developing the search strategy we have decided to not include outcomes as a search terms since it would narrow down the search results too much and exclude other aspects that we aim to extract information about. We believe that the current search strategy will include outcomes even though it is not included as a search term.

Additional search sources:

- The authors state that additional searches for literature will be performed using Google, Google Scholar and ‘possibly other databases as well’. It would be useful if they could add more details about the additional sources. If not planned effectively, website searching can introduce bias.**

Thank you for pointing that out. We have added the following description on page 8:

“Additional searches will be performed manually in Google Search, Google Scholar and relevant databases that collect theses. Overlapping search strategies are necessary when aiming for breadth, and utilizing these search engines increases the likelihood of finding up-to-date grey literature. However, this approach may generate a large number of search hits and the searches may not be easily replicable [cf 29]. Therefore, a systematic approach with defined search terms and a strategy for managing a large volume of search hits will be implemented.”

- It would be useful to see the section that considers the strengths and limitations of each additional search resource.**

Thank you for pointing that out. We have added the following description on page 8:

“Additional searches will be performed manually in Google Search, Google Scholar and relevant databases that collect theses. Overlapping search strategies are necessary when aiming for breadth, and utilizing these search engines increases the likelihood of finding up-to-date grey literature. However, this approach may generate a large number of search hits and the searches may not be easily replicable [cf 29]. Therefore, a systematic approach with

defined search terms and a strategy for managing a large volume of search hits will be implemented.”

9. It might be useful to add searches on online depositories, library catalogues, etc., in addition to Google and Google Scholar – given that the authors' intention is to identify dissertations and theses.

Thank you for pointing that out. We have added the following description on page 8:

“Additional searches will be performed manually in Google Search, Google Scholar and relevant databases that collect theses. Overlapping search strategies are necessary when aiming for breadth, and utilizing these search engines increases the likelihood of finding up-to-date grey literature. However, this approach may generate a large number of search hits and the searches may not be easily replicable [cf 29]. Therefore, a systematic approach with defined search terms and a strategy for managing a large volume of search hits will be implemented.”

10. Do the authors plan to use any software for managing literature reviews, such as COVIDENCE, to aid the review process?

We are planning to use the screening tool Rayyan to aid the review process, as described on page 12:

“The results of the searches from each database will be downloaded in the screening tool Rayyan to support a systematic screening.”

Quality assessment:

11. The authors state that they intend to use the mixed methods appraisal tool to assess the quality of literature included in the review. What about the quality assessment of grey literature? The authors might want to consider using one of the quality assessment tools that have been developed specifically for grey literature – and which are likely to be more appropriate than the mixed methods appraisal tool.

Thank you for this comment, we realise this needs clarification! We will only use grey literature reporting empirical studies and thus we believe that the mixed methods appraisal tool will be useful to assess all of the studies. We have revised and clarified this on page 6:

“For the purpose of this proposed review, grey literature refers to research not intended for publication, or not yet published, in international scholarly journals, i.e. dissertations, thesis, conference publications, unpublished manuscripts, and reports from governmental or non-governmental organizations reporting empirical studies.”

Table 3 on page 10 have been updated with the inclusion criteria regarding type of study:

“Grey literature formed as dissertations, thesis, conference publications, unpublished manuscripts and reports from governmental and non-governmental organizations reporting empirical studies”

Languages other than English (LOE):

12. Including papers in LOE in the review is commendable. Could the authors please state explicitly in their protocol how they are going to search, screen and analyse LOE papers? There is an emerging body of literature that provides strategies for screening LOE titles and abstracts, for example.

We agree. We have already included Swedish, Norwegian and Danish and since we are native Swedish speakers we are also able to read these other Scandinavian languages. This is now clarified on page 11:

"In addition to English, Swedish, Norwegian, and Danish have been chosen for language considerations, as the authors are Swedish and are proficient in these Scandinavian languages."

13. Patient and Public Involvement: In the patient and public involvement section, the authors state that non-governmental organisations have been involved in the development of the review questions. Could the authors please provide more details about this?

We have tried to clarify this on page 15:

"A non-governmental organization advocating the interests of persons experiencing DID, including people with personal experience of DID, have been engaged in the development of the research questions and the design of the review. Members of the board were invited to take part of the research plan and participated in several discussions on the research plan together with the research team. No major changes of the research plan were made after discussions with the non-governmental organization. The findings from the review will be communicated with representatives of the organization."

14. The scoping review methodology allows for the involvement of key stakeholders, including people living with specific health conditions. Given the topic of the review, it would be extremely valuable if the authors considered the involvement of service users and their carers and healthcare providers.

Thank you for your comment, we agree that this is of great value. We have included service users by working with a non-governmental organization advocating the interests of persons experiencing DID, including people with personal experience of DID. Representatives from the board of the organization have been involved in planning the study and the findings will be communicated with them. We have prioritized the view of service users and others advocating the interests of persons experiencing DID since the research team consists of healthcare providers and we believe that we represent that perspective. We have clarified that the non-governmental organization includes persons with own experience of DID on page 15:

A non-governmental organization advocating the interests of persons experiencing DID, including people with personal experience of DID, have been engaged in the development of the research questions and the design of the review. Members of the board were invited to take part of the research plan and participated in several discussions on the research plan together with the research team. No major changes of the research plan were made after discussions with the non-governmental organization. The findings from the review will be communicated with representatives of the organization."